# Research on Coded Excitation Using Kasami Sequence in the Long Rail Detection Based on UGW

Wenqing Yao [1,2], Yuan Yang [1,*] and Xiaoyuan Wei [3]

1   Department of Electronic Engineering, Faculty of Automation and Information Engineering, Xi'an University of Technology, Xi'an 710048, China; yaowenqing@stu.xaut.edu.cn
2   Changzhou Institute of Technology, School of Electrical and Information Engineering, Changzhou 213032, China
3   Department of Electronics Information Engineering, College of Electrical and Information Engineering, Lanzhou University of Technology, Lanzhou 730050, China; weixy@lut.edu.cn
*   Correspondence: yangyuan@xaut.edu.cn

**Abstract:** For a broken rail detection system based on ultrasonic guided waves (UGW), the multi-modal and dispersion characteristics of UGW degrade signal-to-noise ratio (SNR) and range resolution. To improve the SNR of the received signals and range resolution, the coded excitation based on Kasami sequences is presented in this work. Utilizing a PSpice model of piezoelectric ultrasonic transducers, as well as conducting field tests based on the pitch–catch mechanism, it is shown that encoded UGW signals can increase the $SNR_G$ (the gain of SNR) by 6.29 dB. The main lobe width of the coded excitation is mainly determined by the number of carrier cycles and the carrier waveform, and the size of the side lobes is mainly determined by the number of coding bits. To quickly identify the corresponding transmissions at the receivers, a peak detection algorithm is shown. It is based on bandpass filter, triangle filter and Hilbert transform. Its accuracy and effectiveness are validated by using some field tests under different distances. It can be concluded that the shown adaptive peak algorithm has strong robustness and immunity to noise.

**Keywords:** coded excitation; Kasami sequence; ultrasonic guided waves; long rail detection

## 1. Introduction

With the rapid development of high-speed railway, continuous welded rails (CWR) are widely used. Meanwhile, the breakages or defects from temperature stress in the CWR are not to be overlooked. Currently, the detection of rail breakages or defects at home and abroad mainly use ultrasonic flaw detection trolleys, large rail flaw detection vehicles and track circuits. The traditional detection methods mentioned above have some drawbacks such as low detection efficiency, track occupancy, high cost, etc. In recent years, ultrasonic guided waves (UGW) are widely used in various applications, which is resulting from its single point excitation, large detection range, low detection frequency and high detection efficiency. Li et al. [1] used UGW to quantitatively evaluate the debonding between concrete beams and carbon fiber-reinforced polymer (CFRP) overlay. The time-domain spectral element method was used to analyze guided waves in structures for SHM (structural health monitoring) [2]. To impact localization in a stiffened aluminum plate, a laser-based time reversal algorithm was used in [3]. Meo et al. [4] used nonlinear elastic wave spectroscopy to accomplish the damage detection in composite material. To investigate the interaction dynamics of elastic waves with a complex nonlinear scatter, a time reversal mirror is used in [5]. Currently, UGW is also used to detection the rail damage [6–9]. However, in the practical applications, the dispersion and multimode of UGW significantly degrade signal-to-noise ratio (SNR) and range resolution.

In order to improve the SNR of the received UGW signals, some improvements were performed from the transmitting and receiving ends. In the transmitting end, electrical

impedance matching [10], optimal excitation frequency tracking [11], driving circuit optimization [12], and phononic crystals [13,14] were investigated. In the receiving end, some methods of dispersion compensation were involved [15–17]. Xu et al. [15] proposed a guided wave dispersion compensation method by using the compressed sensing technique. Marchi et al. [16] proposed a time-frequency signal processing procedure aimed at extending pulse-echo defect detection methods based on guided waves to irregular waveguides. It is pointed out that the dispersion of UGW can be completely compensated, while the SNR improvement of the received signal is still limited. To further improve SNR, an encoding excitation and pulse compression technique are introduced in UGW-based detection application. This is because encoding excitation can enhance the SNR and guarantee high range resolution at the same time. Fan et al. [18] applied the convolution of Barker and Golay codes as coded excitation signals for low-voltage ultrasonic testing devices. Fu et al. [19] proposed a novel coded excitation by using a linear frequency modulated (LFM) carrier for the purpose of improving ultrasonic imaging quality in terms of axial resolution and SNR. Yucel et al. [20] compared the use of maximal length sequences and LFM signals and proposed a novel signal processing technique by utilizing dispersion compensation and cross-correlation. Wei et al. [21] proposed an adaptive peak detection method to detection breakages of the rail based on Barker coded excitation. Yuan et al. [22] proposed a novel track circuit system based on a previous UGW system and applied 63-bit Kasami sequences to the transmission scheme. To sum up, compared to other codes, Kasami sequences have different code types under the same sequences' length. For the long-range detection system of UGW-based rail breakages, how to quickly distinguish the two receptions from two transmitters is important to detection reliability. So far, different detection frequencies and time intervals were used in two transmitters to identify the two receptions [23]. This will highly limit the large-scale application of UGW-based broken rail detection technique. Thence, to quickly and accurately distinguish two receptions, the investigation of the encoding excitation technique based on Kasami sequences used in detection of the long rail is urgently required.

Given these considerations mentioned above, the encoding excitation based on Kasami sequences and the corresponding peaks detection are investigated in this paper by using the established PSpice model of sandwiched piezoelectric ultrasonic transducers (SPUT). It can be concluded that the encoding excitation based on Kasami sequences can effectively improve SNR, and the presented peak detection algorithm can rapidly identify peaks. The rest of this paper is organized as follows: Section 2 illustrates the PSpice model of SPUT in longitudinal vibration as well as the scheme of Kasami sequences encoding and decoding; Section 3 presents the encoding excitation analyzed results and peaks detection algorithm. The discussion is present in Section 4; finally, some conclusions are given in Section 5.

## 2. Materials and Methods

### 2.1. PSpice Model of SPUT in Longitudinal Vibration

The structure diagram of an SPUT in longitudinal vibration is shown in Figure 1. From Figure 1, it mainly includes front mass, back mass, metal electrodes and a piezoelectric ceramic stack with a prestressed bolt. By using one-dimensional wave and cascade theory, the PSpice model of SPUTs in longitudinal vibration was established in [24], as shown in Figure 2. The specific illustration of the PSpice model of SPUT was given in [24]. It should be noted that the effect of a prestressed bolt on the electromechanical characteristics of SPUT cannot be considered in the established PSpice model of SPUT. The metal electrodes are modeled by means of the lossless transmission line. The parameters of the lossless transmission line can be calculated by the following Equations (1) and (2) [25,26]:

$$F = \frac{NL}{LEN} v_t \tag{1}$$

$$Z_0 = \rho v_t A \tag{2}$$

where $F$ represents resonance frequency, $LEN$ indicates the length of the transmission line, $NL$ denotes the normalized length, $v_t$ represents sound velocity, $Z_0$ represents the characteristic impedance, $\rho$ is the material density, and $A$ indicates the cross-sectional area. The density and sound speed in air are given as $\rho_1 = 1.293 \, \text{kg/m}^3$, $v_1 = 340 \, \text{m/s}$, and the cross-sectional area of the back mass of SPUT is indicated by $A_1 = 1075.21 \, \text{mm}^2$. Substituting these parameters into Equation (2), the equivalent air load can be calculated as $0.472 \, \Omega$.

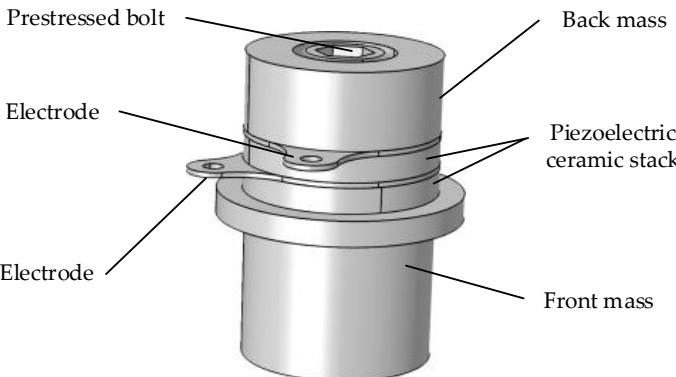

**Figure 1.** Structure diagram of an SPUT in longitudinal vibration.

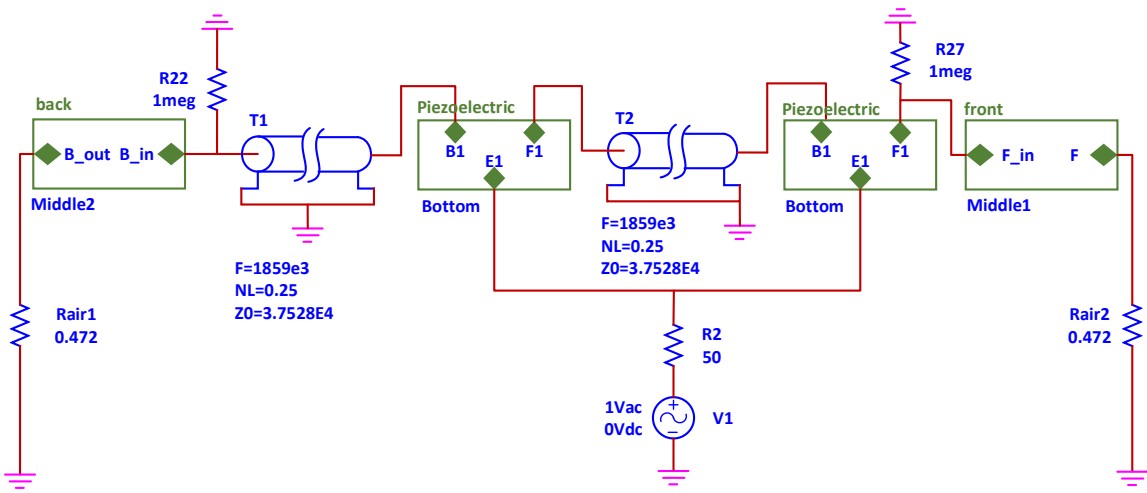

**Figure 2.** PSpice model of an SPUT in longitudinal vibration.

### 2.2. Scheme of Kasami Sequences Encoding and Decoding

Kasami sequences include a small set and a large set, which stems from m-sequences. The detailed procedure was explained in [27]. The small set of Kasami sequences is chosen in this paper, resulting from its better cross-correlation features than that of the large set. In general, the small set can be expressed as $Kas(n, M, L)$, where $n$ is a non-negative even integer, $M = 2^{n/2}$ indicates the number of sequences contained in the set, and $L = 2^n - 1$ is the length of each sequence in the set. In the case of periodic transmission of Kasami sequences, auto-correlation functions (ACF) and cross-correlation functions (CCF) can be represented as:

$$ACF(\tau) = \begin{cases} L, & \tau = 0 \\ \{-1, -M-1, M-1\}, & \tau \neq 0 \end{cases} \qquad (3)$$

$$CCF(\tau) = \{-1, -M-1, M-1\} \qquad (4)$$

where $L$ represents the length of each sequence, and $M$ indicates the number of sequences.

Taking 63-bit Kasami sequences as an example, the corresponding ACF and CCF are calculated, as shown in Figure 3. From Figure 3, it can be found that Kasami sequences have excellent ACF and low CCF characteristics. It should be noted that good ACF is very helpful for the identification of correlation peaks at the receiver, while the low CCF values will more effectively avoid misunderstanding other maximum values from some noises or other transmissions with the correct correlation peaks [22]. In fact, 63-bit Kasami sequences have eight types, while 15-bit Kasami sequences also have four types. The Kasami sequences used in this paper are listed in Table 1.

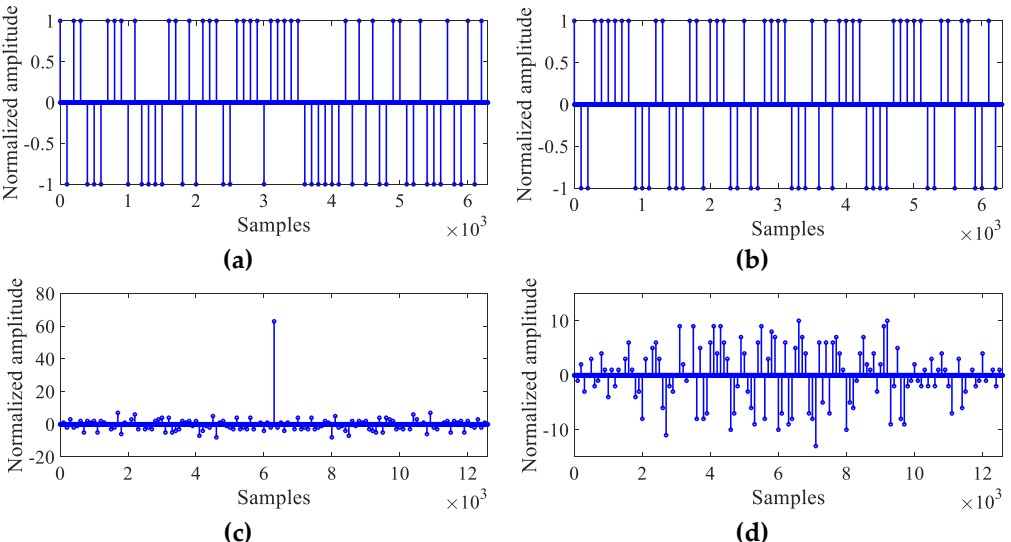

**Figure 3.** Kasami sequence analysis of ACF and CCF, (**a**) one 63−bit Kasami sequence, (**b**) the other 63−bit Kasami sequence, (**c**,**d**) the corresponding ACF and CCF.

**Table 1.** Kasami sequences used in this paper.

| Bits | Types | Specific Values |
|---|---|---|
| 63 | 01 | $1 \to -1 \to 1 \to 1 \to -1 \to -1 \to -1 \to 1 \to 1 \to 1 \to -1$ $\to 1 \to -1 \to -1 \to -1 \to -1 \to 1 \to 1 \to -1 \to 1 \to -1$ $\to 1 \to 1 \to 1 \to -1 \to -1 \to 1 \to 1 \to 1 \to 1 \to -1 \to 1 \to$ $1 \to 1 \to 1 \to 1 \to -1 \to -1 \to -1 \to -1 \to -1 \to -1 \to 1$ $\to -1 \to 1 \to -1 \to 1 \to -1 \to -1 \to 1 \to 1 \to -1 \to -$ $1 \to 1 \to -1 \to -1 \to -1 \to 1 \to -1 \to -1 \to 1 \to -1 \to 1$ |
| 63 | 02 | $1 \to -1 \to -1 \to 1 \to 1 \to 1 \to 1 \to 1 \to 1 \to -1 \to -1 \to$ $-1 \to 1 \to 1 \to -1 \to -1 \to -1 \to 1 \to 1 \to -1 \to 1 \to 1$ $\to 1 \to -1 \to -1 \to 1 \to -1 \to -1 \to 1 \to 1 \to 1 \to 1 \to$ $-1 \to -1 \to -1 \to 1 \to -1 \to 1 \to -1 \to 1 \to 1 \to 1 \to 1$ $\to -1 \to -1 \to -1 \to -1 \to 1 \to 1 \to 1 \to 1 \to 1 \to -1 \to$ $-1 \to 1 \to 1 \to -1 \to 1 \to 1 \to -1 \to -1 \to 1 \to -1$ |
| 15 | 01 | $-1 \to -1 \to 1 \to 1 \to -1 \to 1 \to 1 \to 1 \to -1 \to -1 \to -1$ $\to -1 \to 1 \to -1 \to 1$ |

The baseband spectrum of Kasami sequences cannot satisfy the bandwidth of SPUTs used in this paper. Therefore, Kasami sequences cannot be used directly to excite an SPUT, which is the same as other codes. Then, the BPSK (Binary Phase Shift Keying) technique is used to modulate the Kasami sequences with a carrier signal. The transmitted signal $e[k]$ can be given as [22]

$$e[k] = \sum_{i=0}^{N_c \cdot N_s \cdot L - 1} c\left[\frac{i}{N_c \cdot N_s}\right] \cdot p[k - i] \tag{5}$$

where $c[k]$ represents the Kasami sequence used to encode the excitation signal, $p[k]$ denotes a modulation symbol composed of $N_c$ cycles of the selected carrier, $N_s$ indicates the sampled number of a carrier period, and $L$ indicates the length of Kasami sequence. Before performing the correlation operation, firstly, the signal $y[k]$ acquired by the receiver is carried out with BPSK demodulation. The demodulated signal $y_d[k]$ can be given as

$$y_d[k] = \sum_{i=0}^{N_c \cdot N_s - 1} y[i+k] \cdot p[i] \tag{6}$$

subsequently, the detection of the encoding transmission can be accomplished by using correlation operation.

$$t[k] = \sum_{i=0}^{L-1} y_d[i \cdot N_c \cdot N_s + k] \cdot c[i] \tag{7}$$

where $t[k]$ represents the correlation output, and $c[i]$ represents the Kasami sequence used to encode the excitation signal. The peak detection algorithm is used to analyze the correlation signal, which is used to detect the arrival of transmission signals.

## 3. Results

### 3.1. Analysis of Encoding Excitation by Kasami Sequences

3.1.1. Comparison Analysis of Single Pulse and Encoding Excitation

To validate the feasibility, the effect of single pulse and encoding excitation on the transient characteristics of SPUTs is analyzed based on the PSpice model from Figure 2. The single pulse is sinusoidal signal, and its frequency and cycle numbers are 35 kHz and 5, respectively. The encoding excitation selects 63-bit Kasami sequences, and the corresponding carrier is the single pulse. The vibration velocity of the front mass is chosen to reflect the transient characteristics of SPUTs; it can be obtained by using the voltage probe in Pspice software. The specific analyzed results are shown in Figure 4.

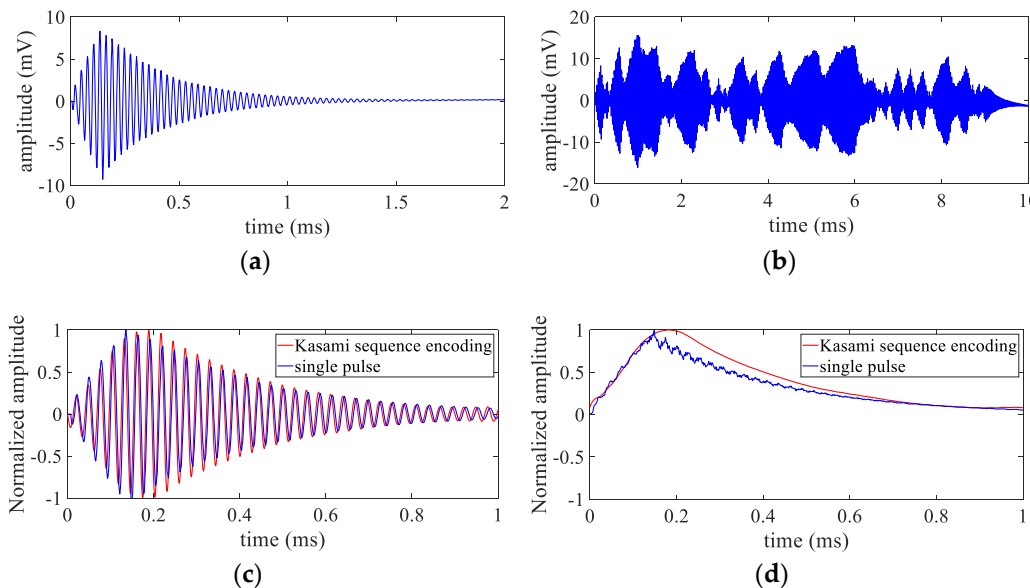

**Figure 4.** Comparison analysis of single pulse and Kasami sequence encoding excitation, (**a**) vibration signal under single pulse, (**b**) vibration signal under Kasami sequence encoding, (**c**,**d**) normalized waveform and envelope.

Under the premise that the noise is determined, the gain of SNR is defined as:

$$SNR_G = 10 \cdot \log(\frac{A_{s1}}{A_{s2}}) \tag{8}$$

where $SNR_G$ represents the gain of SNR, and $A_{s1}$ and $A_{s2}$ indicate the amplitude of the received signals. From Figure 4a,b, it is shown that the amplitude of the vibration signal in the front mass is much larger than that of single pulse: about twice as much. According to Figure 4a,b, $A_{s1}$ and $A_{s2}$ can be obtained as 8.30 mV, 15.57 mV. Substituting these parameters into Equation (8), the $SNR_G$ can be obtained as 6.29 dB. Considering Figure 4c,d, after performing amplitude normalization, the waveforms from encoding excitation and single pulse are almost the same. There are subtle differences; the change trend is consistent according to the extracted envelopes under the two excitation signals above. A conclusion can be drawn that the encoding excitation based on Kasami sequences can be used to generate UGW, which is suitable for the rail detection.

3.1.2. The Effect of Carrier Signals and Bits of Kasami Sequences on Encoding Excitation

To select appropriate bits and carrier cycles of Kasami sequences, the ACF characteristics of Kasami sequences are analyzed. Here, the carrier signal is a five-cycle sinusoidal signal and its frequency is 35 kHz. Then, the normalized ACF envelopes of 15-bit and 63-bit Kasami sequences are calculated, as shown in Figure 5a,b. From Figure 5a, compared to a 15-bit Kasami sequence, the corresponding envelope of a 63-bit Kasami sequence has smaller side lobes. From Figure 5b, the main lobe width of the two envelopes is almost the same.

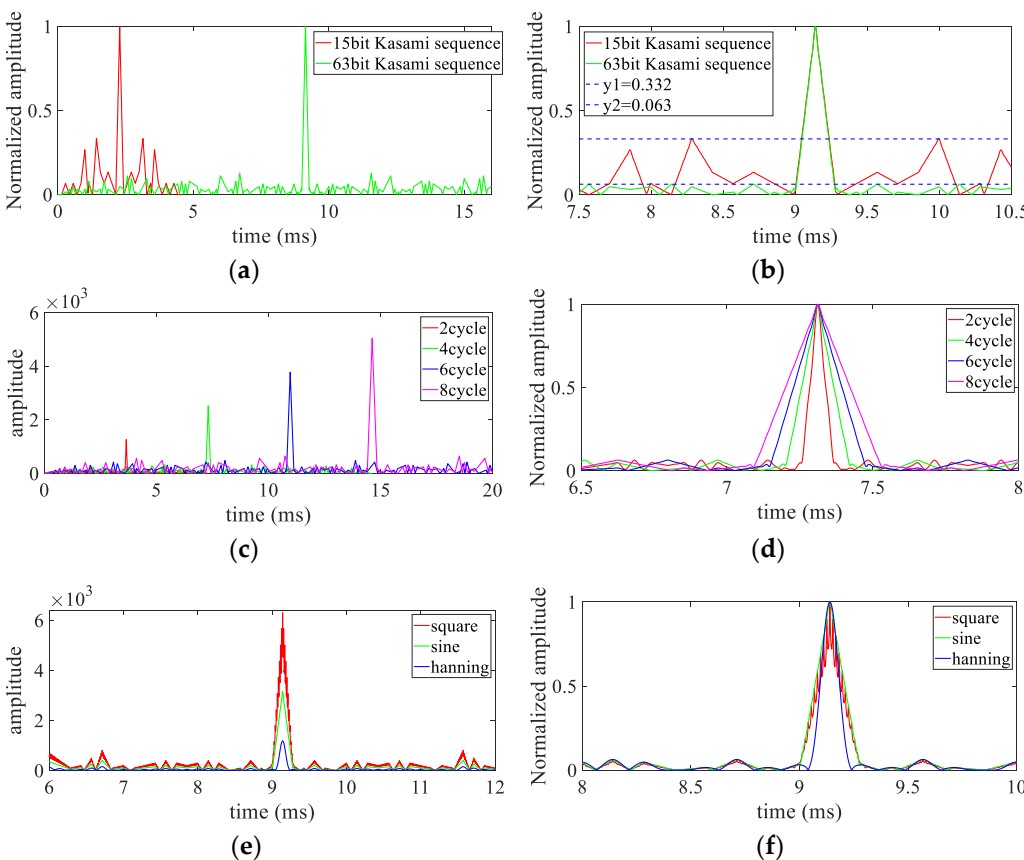

**Figure 5.** ACF envelopes analysis of Kasami sequences, (**a,b**) comparing the effect of different bits on ACF envelopes, (**c,d**) comparing the effect of different cycles on ACF envelopes, (**e,f**) comparing the effect of different carrier waveform.

For the UGW inspection system based on coding excitation, there are two important characteristic parameters used for evaluating UGW detection signals: Main Lobe Width (MLW, Main Lobe Width) and Sidelobe Peak Level (PSL, Peak Sidelobe Level). The PSL formula in decibels can be given as [21]

$$PSL = 10 \log \left[ \frac{\max(A_{sidelobe})}{\max(A_{mainlobe})} \right] \tag{9}$$

where $A_{sidelobe}$ and $A_{mainlobe}$ indicate peak values of sidelobes and main lobes, respectively. A straight line parallel to the time axis is drawn when the amplitude of the envelope signal drops to 6 dB. Then, the projected line segment on the time axis after the line intersects the envelope signal represents MLW; it can be given as [21]

$$MLW = t_2 - t_1 \tag{10}$$

MLW is used as the basis for whether the detection signal can be distinguished. When the time delay of the two signals $|t_d| \leq MLW$, the two signals are difficult to distinguish. Only when $|t_d| > MLW$ can the two signals can be distinguished. According to Figure 5b, the corresponding PSL and MLW under different bits Kasami sequences can be calculated as:

$$
\begin{aligned}
15bit \ \text{PSL} : \ &10 \log(0.332/1) = -11.03 \ \text{dB} \quad 15bit \ \text{MLW} : \ 9.221 - 9.068 = 0.143 \ \text{ms} \\
63bit \ \text{PSL} : \ &10 \log(0.063/1) = -27.65 \ \text{dB} \quad \ 63bit \ \text{MLW} : \ 9.21 - 9.07 = 0.14 \ \text{ms}
\end{aligned} \tag{11}
$$

From Equation (11), it can be inferred that in the case of the same carrier, the more bits of the sequence, the smaller the side lobes, but the width of the main lobe is almost unchanged. From this point of view, in order to obtain a lower sidelobe level, a Kasami sequence with a larger length is required.

Based on the above analysis conclusions, it is easy to think whether the envelope width of the autocorrelation function of the Kasami sequence is closely related to the number of cycles of the carrier signal. Here, taking the 63-bit Kasami sequence as an example, the carrier signal adopts a sinusoidal signal with a frequency of 35 kHz, and the number of cycles is 2, 4, 6 and 8 cycles, respectively. The autocorrelation function envelopes of the Kasami sequence with different carrier cycles is analyzed, as shown in Figure 5c,d. From Figure 5c, as the number of carrier cycles increases, the peak value of the autocorrelation function envelope increases linearly. From Figure 5d, after normalizing the amplitude of the envelope, it is found that as the number of carrier cycles increases, the main lobe width of the autocorrelation function increases linearly. Therefore, under the premise that the carrier waveform is determined, the main lobe width of the autocorrelation function of the Kasami sequence is only related to the number of carrier cycles.

To analyze the influence of different carrier waveforms on the envelope of the autocorrelation function, this paper uses square waves, sinusoidal signals and sinusoidal signals modulated by the Hanning window as carrier signals. These signals are with a frequency of 35 kHz and five cycles. The specific analyzed results are shown in Figure 5e,f. From Figure 5e, it can be seen that the envelope peak value of the autocorrelation function corresponding to the square-wave signal under the same amplitude condition is the largest, which is followed by the sinusoidal signal, and the envelope peak value corresponding to the sinusoidal signal modulated by the Hanning window is the smallest. From Figure 5f, after the amplitude normalization of the above envelope, it is found that the main lobe width of the envelope corresponding to the sinusoidal signal modulated by the Hanning window is the smallest, and the main lobe width of the envelope corresponding to the square wave and the sinusoidal signal is approximately the same. To sum up, the main lobe width of the autocorrelation function envelope of the coded excitation is mainly determined by the number of carrier cycles and the carrier waveform, and the size of the side lobes is mainly determined by the number of coding bits.

### 3.1.3. Field Test of Kasami Sequences on Encoding Excitation

In order to verify the correctness of the above analysis results, an experimental platform based on the pitch–catch mechanism is built, as shown in Figure 6. The test uses 100 m and 200 m CHN60 rails, and the carrier signal selects a square wave pulse with a frequency of 35 kHz. From Figure 6, E1 represents the transmitting transducer, while R1 and R2 indicate the receiving transducer, respectively.

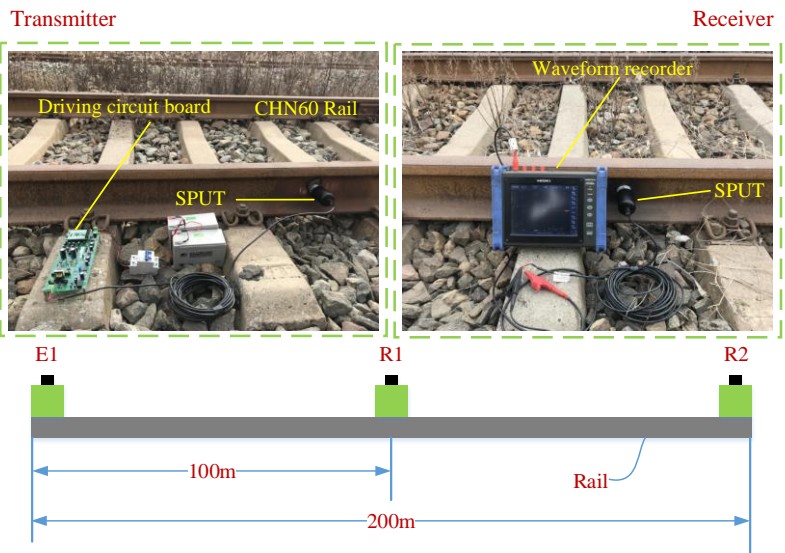

**Figure 6.** Field test diagram of encoding excitation.

Based on the above experimental platform, the ultrasonic excitation signal encoded by the 15 bit Kasami sequence is tested in the rail with a length of 100 m, and the number of carrier cycles is selected as 2, 4, 6 and 8, respectively. According to Figure 7a–d, it can be seen that with the increase in the carrier cycle number, the received signal amplitude firstly increases, and then, it is almost unchanged. From Figure 7e–h, it is easy to find that as the number of carrier cycles increases, the main lobe width of the decoded signal envelope gradually widens, indicating that the main lobe width of the decoded signal envelope is directly related to the number of carrier cycles, which is consistent with the conclusion obtained by analyzing the main lobe of the autocorrelation function envelope of the encoding signal by Kasami sequence.

Likewise, the ultrasonic excitation signal encoded by the 63-bit Kasami sequence is tested in the rail with a length of 100 m based on the above experimental platform, and the number of carrier cycles is selected as 1, 2, 3 and 4, respectively. From Figure 8a–d, it can be seen that as the number of carrier cycles increases, the amplitude of the received signal increases gradually at first, and then it shows a downward trend. According to Figure 8e–h, it is easy to find that as the number of carrier cycles increases, the main lobe width of the decoded signal envelope gradually widens, which is consistent with the conclusion obtained in the Section 3.1.2.

From Figures 7 and 8, the values of the received signal amplitude, MLW and PSL under different bits and carrier cycles are obtained. The effect of carrier cycles on the amplitude, MLW and PSL is analyzed as shown in Figure 9. From Figure 9a,b, with the increasing of carrier cycles, the amplitude values are gradually increased and then almost unchanged, while the MLW are gradually improved. From Figure 9c,d, with the increasing of carrier cycles, the amplitude values first increase and then gradually decrease, while the corresponding MLW are gradually improved. Combined Figure 9e,f, it is found that with the increasing of carrier cycles, PSL presents minor changes. To summarize, the MLW is closely related with carrier cycles when Kasami sequences bits are unchanged, and the number of carrier cycles has little effect on PSL.

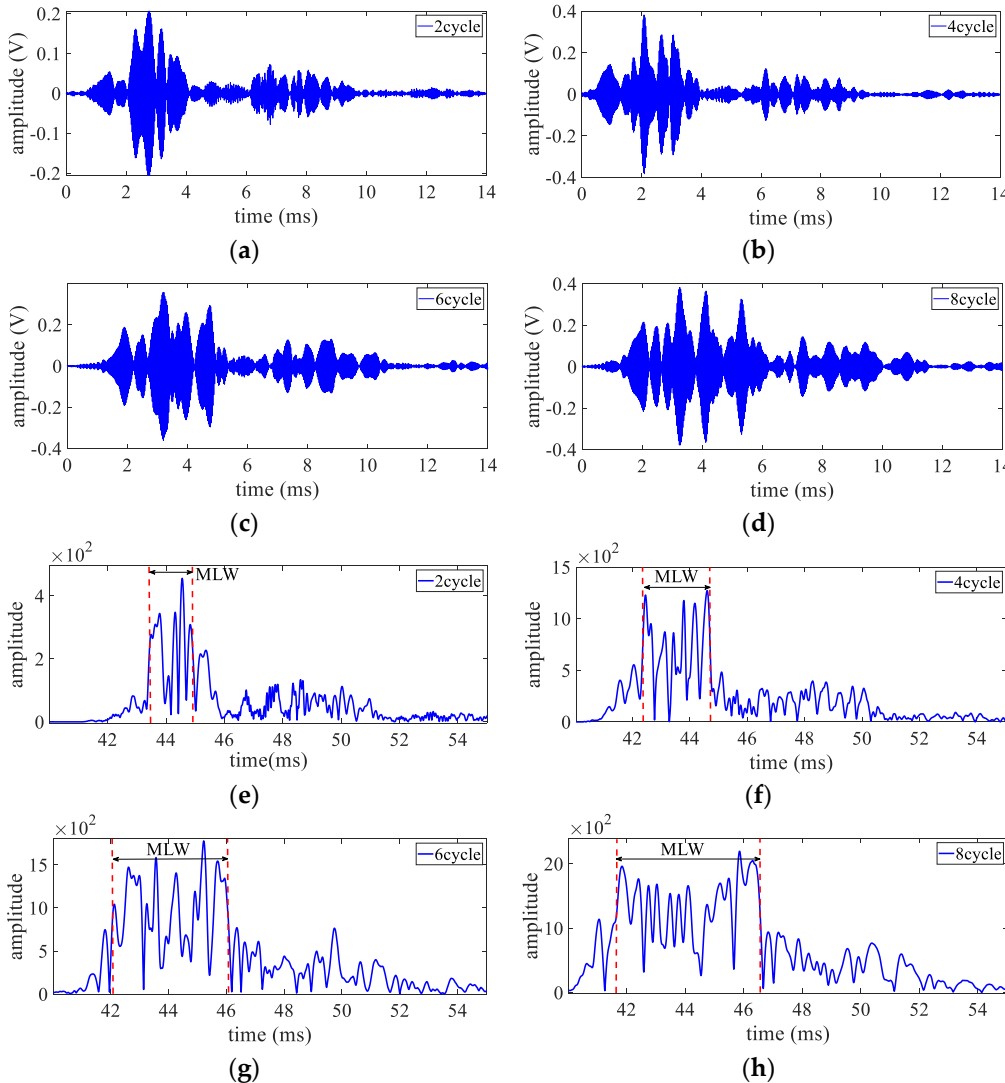

**Figure 7.** Results analysis of encoding excitation by 15−bit Kasami sequence under different carrier cycles, (**a**–**d**) the corresponding received signals, (**e**–**h**) the decoding results.

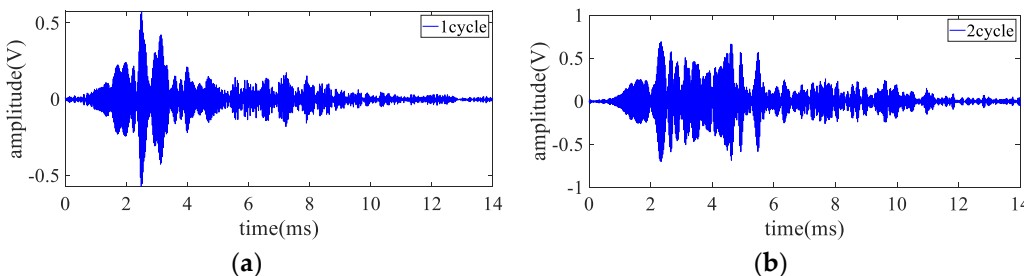

**Figure 8.** *Cont.*

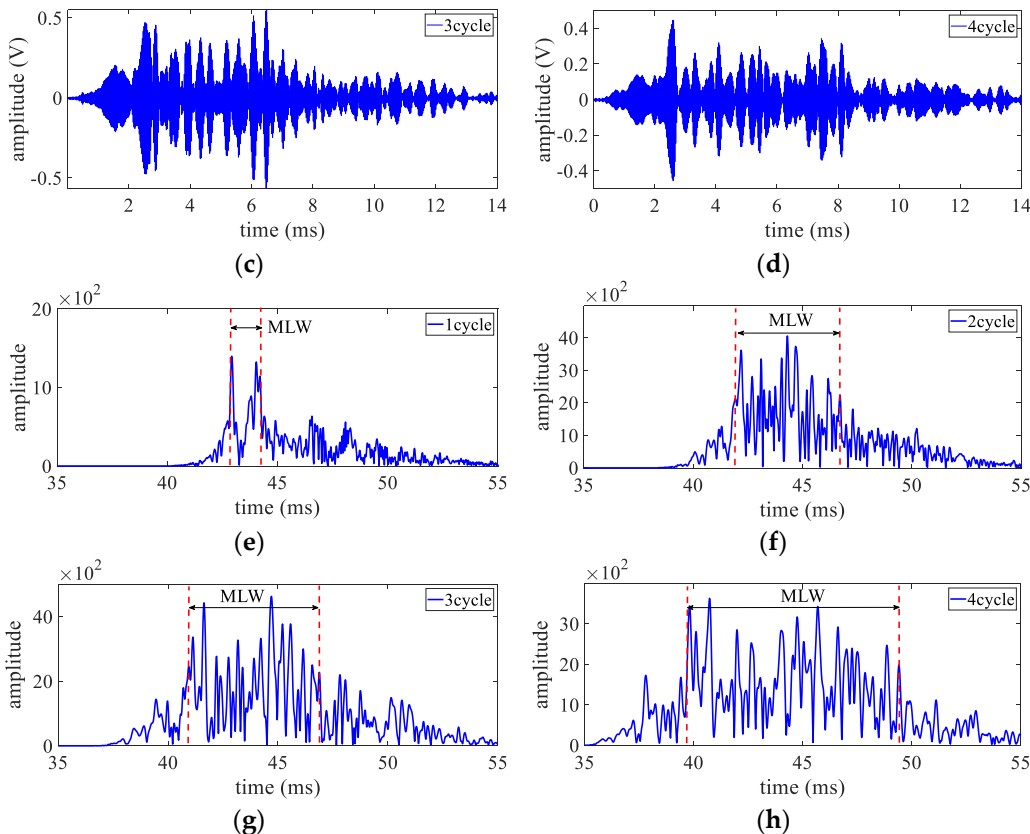

**Figure 8.** Results analysis of encoding excitation by 63−bit Kasami sequence under different carrier cycles, (**a**–**d**) the corresponding received signals, (**e**–**h**) the corresponding decoding results.

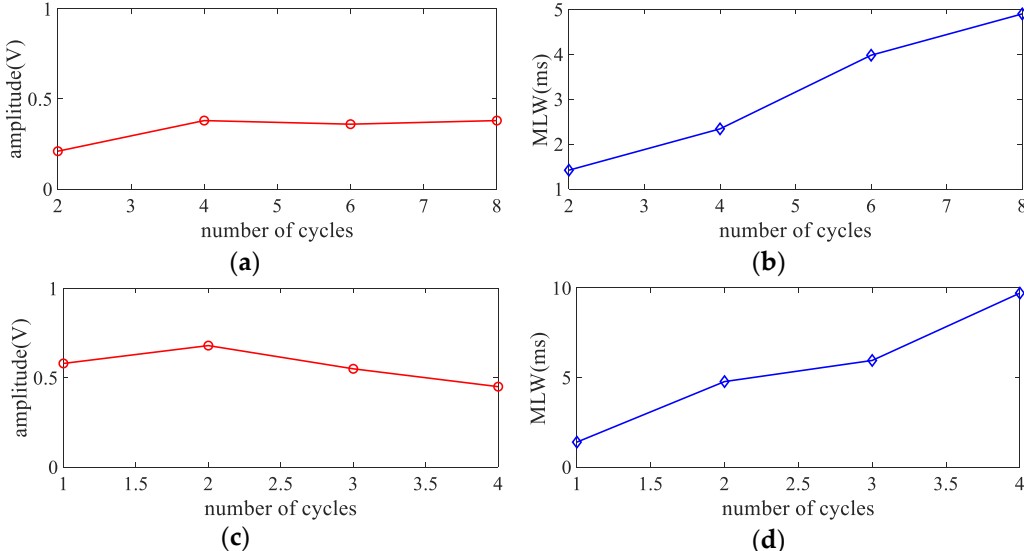

**Figure 9.** *Cont*.

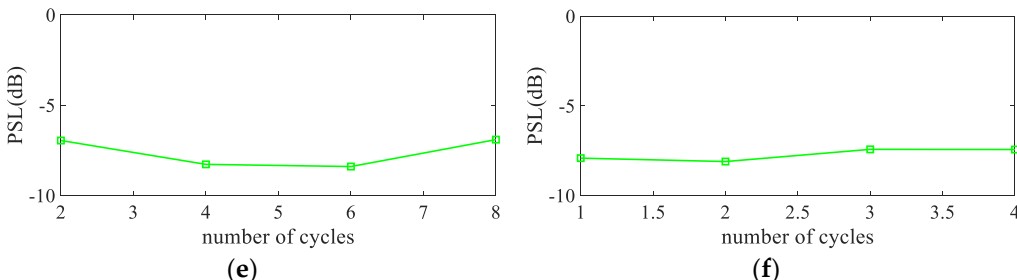

**Figure 9.** The effect analysis of carrier cycles on the amplitude of received signals and MLW, (**a**) 15-bit amplitude, (**b**) 15−bit MLW, (**c**) 63−bit amplitude, (**d**) 63−bit MLW, (**e**) 15−bit PSL, (**f**) 63−bit PSL.

### 3.2. Peaks Detection

According to the analysis in Section 3.1, in the UGW detection system based on the pitch–catch mechanism, after the SPUT is excited by the encoding excitation, the characteristics of the received signal by SPUTs at the receiving end are not easily obtained, especially in the determining of peaks arrival time. Generally, the premise of rail damage location is an accurate measurement of flight time; the measurement of flight time based on peak value is one of the common and effective methods. Firstly, taking a 63-bit Kasami sequence with four carrier cycles as an example, the received signals under different detection distances are analyzed, as shown in Figure 10. From Figure 10a,d, it can be seen that when the test distance is doubled, the signal length is doubled but the signal amplitude is significantly reduced, which is closely related to the dispersion characteristics of UGW in the rail.

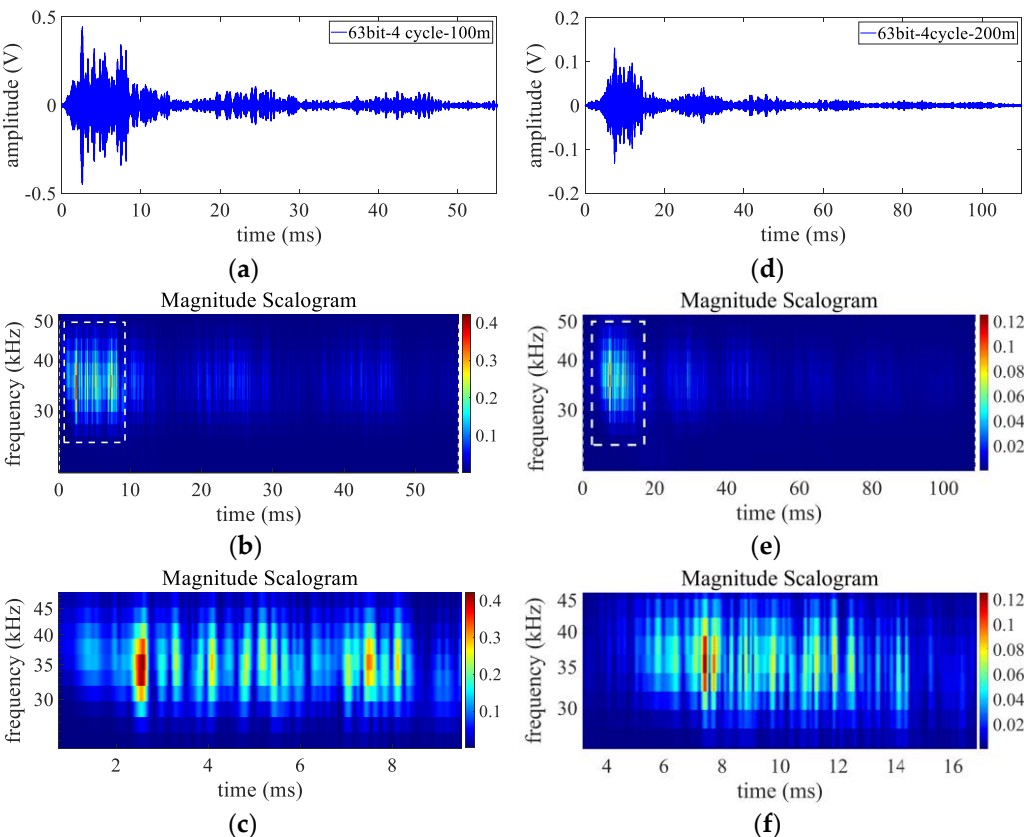

**Figure 10.** The received signals under encoding excitation by 63−bit Kasami sequence, (**a**) the received signal under test distance equaling 100 m, (**b**,**c**) the corresponding time frequency analysis results, (**d**) the received signal under test distance equaling 200 m, (**e**,**f**) the corresponding time frequency analysis results.

Subsequently, in order to analyze the time-frequency characteristics of the received signal above, continuous wavelet transform is used, and the specific analysis results are shown in Figure 10. From Figure 10c,f, it is shown that the energy of the received signal is mainly concentrated in the range of 30 to 45 kHz, where the energy is stronger in the range of 32.5 to 37.5 kHz. From the perspective of time, it mainly focuses on 0~15 ms, so we focus on analyzing the first 15 ms signal. When the sampling rate is 500 kHz, it is enough to analyze 7500 sampling points. Hence, the bandwidth of the bandpass filter in the peak detection algorithm is to be selected as 5 kHz (32.5–37.5 kHz), and the center frequency is 35 kHz.

The Hilbert transform is widely used in analyzing the instantaneous amplitude and frequency of signals, and its physical meaning is essentially an all-pass network with a phase lag $\pi/2$. By its physical meaning, it is easy to find that after the Hilbert transform of the envelope signal, the peak of the envelope corresponds to the forward over-zero point of the transformed signal, so the Hilbert transform is used in this paper to achieve effective detection of the peak of the envelope. The Hilbert transform of a real signal is defined as follows [28]:

$$\hat{x}(t) = H[x(t)] = \frac{1}{\pi t} * x(t) = \frac{1}{\pi}\int_{-\infty}^{\infty}\frac{x(\tau)}{t-\tau}d\tau \tag{12}$$

where $*$ represents convolution operators. From Equation (12), the Hilbert transform of a signal $x(t)$ is essentially its convolution with the $1/\pi t$. The frequency domain form of the Hilbert transform can be defined as follows:

$$\hat{X}(t) = F\left[\frac{1}{\pi t}\right] \cdot F[x(t)] = -j\mathrm{sgn}(f)X(f) \tag{13}$$

where $X(f)$ is the Fourier transform of the signal $x(t)$ and $j$ is the imaginary unit.

The peak detection logic based on Hilbert transform is shown in Figure 10. From Figure 11, to obtain the one-to-one correspondence between the envelope peak point and the positive zero-crossing point of the Hilbert transformed signal, it is necessary to ensure that the envelope signal is very smooth. Therefore, before peak detection, it is necessary to preprocess the detection signal at the receiving end to obtain a smooth envelope signal.

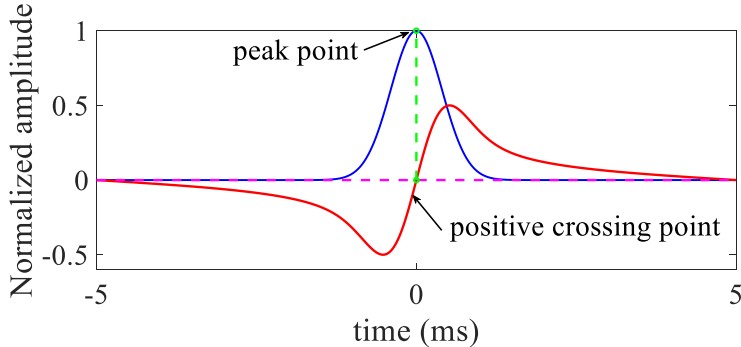

**Figure 11.** Peaks detection logic.

The processing flow of the adaptive peak detection algorithm given in this paper is shown in Figure 11, which mainly involves bandpass filtering, amplitude normalization, square transform, triangular filtering, envelope extraction and Hilbert transform. For the BPSK demodulation and encoding signal detection shown in Figure 12, the above processing can be realized according to Equations (6) and (7).

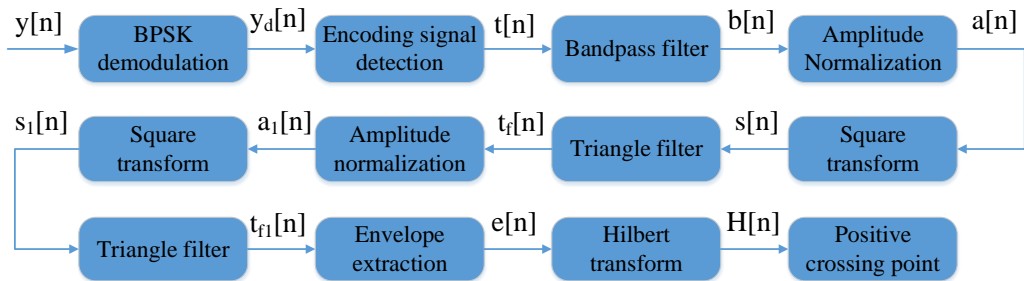

**Figure 12.** Peak detection algorithm flow.

The amplitude normalization of the bandpass filtered signal $b[n]$ can be given as:

$$a[n] = norm(b[n]) = \frac{b[n]}{\max(b[n])} \tag{14}$$

to simplify the operation and to highlight the peaks of detected signal, the amplitude normalized signal $a[n]$ is squared by Equation (15).

$$s[n] = a[n] \cdot a[n] \tag{15}$$

after the above square transform nonlinear processing, the envelope is extracted from the signal $s[n]$ by a triangular filter $t_f[n]$. In fact, the triangular filter is a low-pass filter, and it consists of a two-stage moving average filter $m[n]$ cascade and can be expressed as [21]:

$$t_f[n] = m[n] * m[n] \tag{16}$$

$$m[n] = \frac{1}{N_w}[s(n - (N_w - 1)) + s(n - (N_w - 2)) + \cdots + s(n)] \tag{17}$$

where $N_w$ represents the window width. It should be pointed out that the window width is generally equal to the main lobe width of $b[n]$. Then, the smooth envelope signal is obtained by processing according to Equations (14)–(16), and the Hilbert transform is applied to obtain a unique positive over-zero point to identify the peaks.

To verify the correctness of the peak detection method presented in this paper, the signals from Figure 10a,d are analyzed, and the analyzed results are shown in Figure 13. From Figure 13, the peak detection method presented in this paper without any threshold can accurately extract the peak of the received signal under different test distances. Compared to Figure 13c, the envelope of the detection signal from Figure 13g becomes wider, which is related to the dispersion effect of the UGW in the rail.

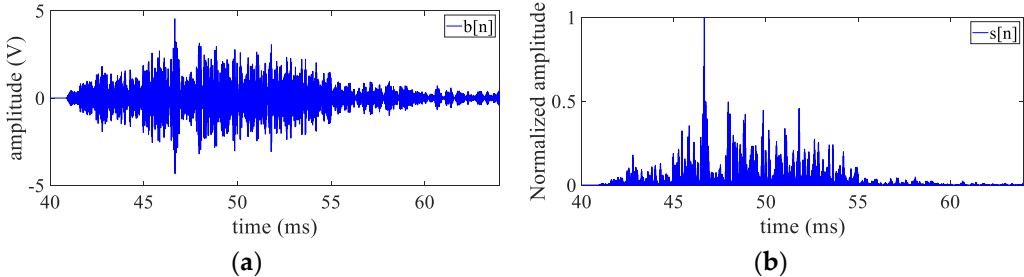

**Figure 13.** *Cont.*

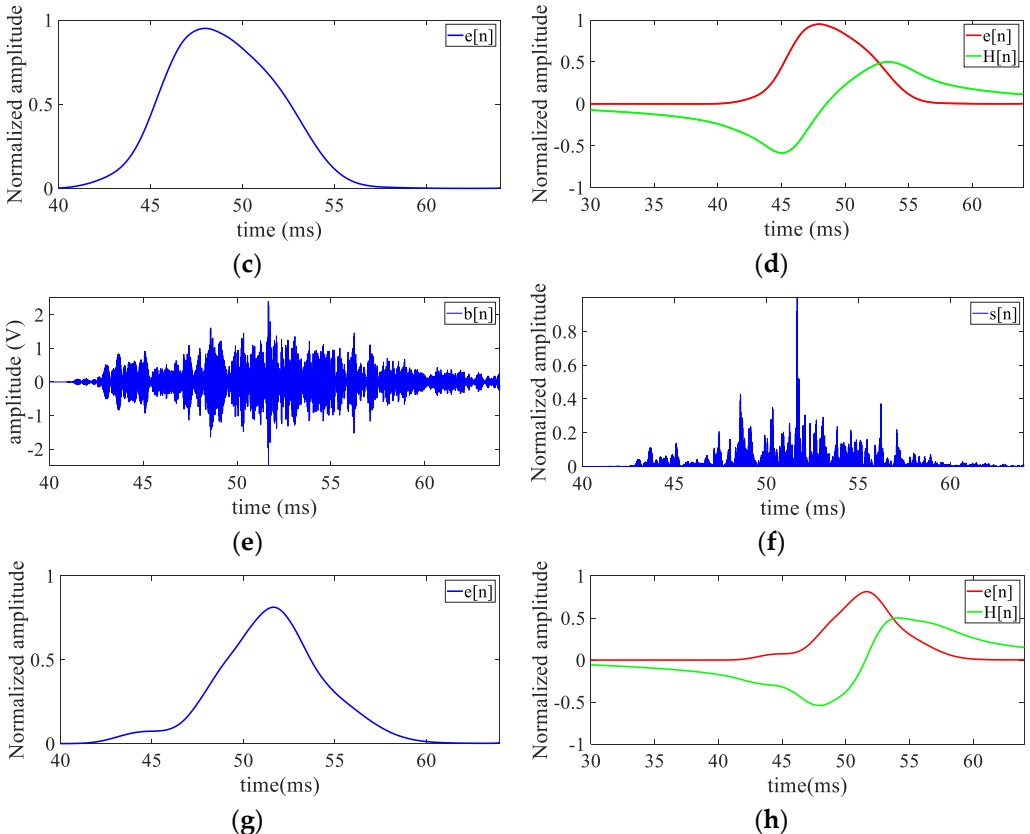

**Figure 13.** Peak detection of the received signals under different test distances, (**a**) bandpass filtered signal under test distance equaling 100 m, (**b**) the corresponding square transformed signal, (**c**) the corresponding envelope, (**d**) peaks detection results, (**e**) bandpass filtered signal under test distance equaling 200 m, (**f**) the corresponding square transformed signal, (**g**) the corresponding envelope, (**h**) peaks detection results.

Subsequently, to analyze the anti-noise performance of the peak detection algorithm presented in this paper, Gaussian white noise with SNR = −20 dB is superimposed for the received signal at the test distance of 200 m, as shown in Figure 14. From Figure 14, it can be found that the characteristics of the received signal have been completely drowned by the noise, but after the peak detection process, the only positive crossing point corresponding to the envelope peak can still be obtained, which fully illustrates that the peak detection method presented in this paper has strong anti-noise performance and meets the requirements of in-service rail UGW detection.

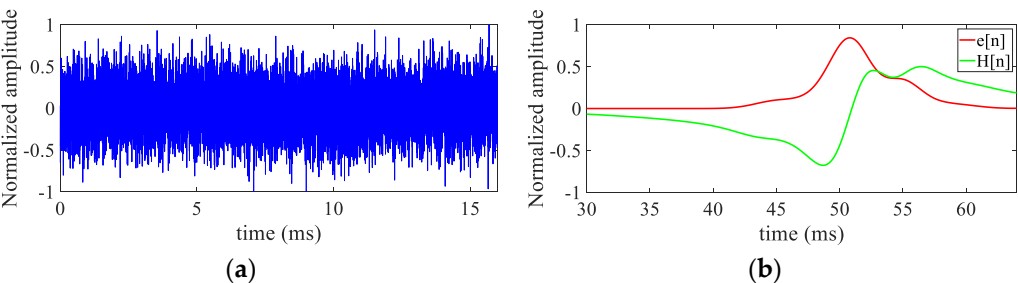

**Figure 14.** The analysis of peak detection under Gaussian noise, (**a**) SNR = −20 dB, (**b**) peaks detection.

## 4. Discussion

The encoding excitation based on Kasami sequences and the corresponding adaptive peak detection algorithm are presented in this work. Firstly, the encoding excitation used

in detection of the long rail is investigated. The effect of Kasami sequence bits, carrier waveform and cycles on encoding excitation is analyzed. From a theoretical perspective, the more Kasami sequence bits, the less PSL and more MLW. However, on the basis of the analyzed results shown in Section 3.1, the performance of encoding excitation failed to improve when the Kasami sequence bits are increased under carrier waveform and cycles remaining unchanged. It is inferred that the performance difference is resulting from UGW dispersion in the rail. The corresponding dispersion compensation is performed before decoding the encoding signal. Secondly, the adaptive peak detection algorithm based on bandpass filter, triangle filter and Hilbert transform is shown. The algorithm is validated by using the received signal from a field test under different test distances. It should be noted that the corresponding time of the positive crossing point is not the actual peak time. Taking a 15-bit Kasami sequence as an example, the specific analyzed process is shown in Figure 15. From Figure 15, the time difference mainly is from BPSK demodulation, encoding signal detection, bandpass filtering, and triangle filtering. For Kasami sequences, BPSK demodulation and encoding signal detection are accomplished by cross-correlation. The cross-correlation function in MATLAB software is given as:

$$r = xcorr(x, y) \qquad (18)$$

where $x$ and $y$ represent discrete time series. If $N$ is the length of the longer of $x$ and $y$, the length of the discrete time series $r$ can be calculated by $2N - 1$. According to Equation (18), the cross-correlation will generate the time shift. In addition, the time shift from bandpass filtering, triangle filtering and Hilbert transform is different for different received signals. In order to improve defect positioning accuracy, the above time shift should be small as soon as possible. Finally, from the pitch–catch mechanism, the encoding excitation based on the Kasami sequence is discussed. In fact, two typical situations including transmitter–receiver– transmitter (TRT) and receiver–transmitter–receiver (RTR) are present in the real-time online detection system based on UGW. The encoding excitation under TRT and how to reduce the above time shift will be expected to performed in subsequent research work.

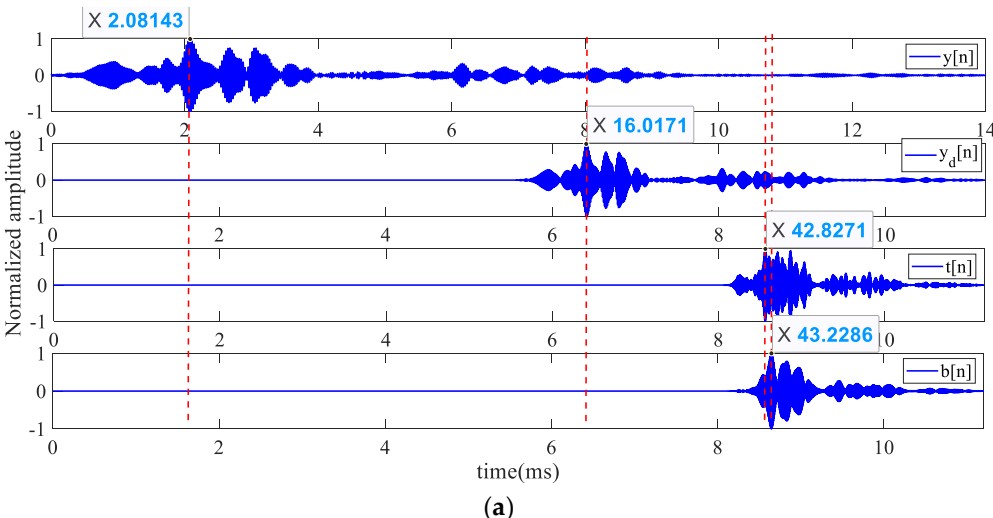

(a)

**Figure 15.** *Cont.*

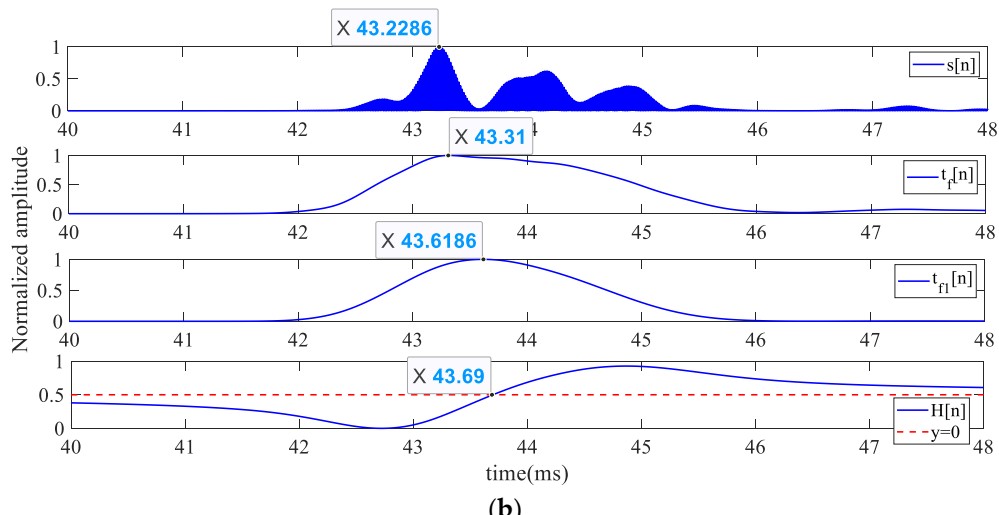

(**b**)

**Figure 15.** The analysis of peak detection time under test distance equaling to 100 m, (**a**) encoded excitation decoding, (**b**) peak detection.

## 5. Conclusions

According to the pitch–catch mechanism, which is similar to RTR, the effect of Kasami sequence bits, carrier waveform and cycles on encoding excitation, and Kasami sequence coded UGW signals are analyzed in this work by using simulations and field tests. From the aforementioned coded transmissions, a peak detection algorithm is presented, which is based on bandpass filter, triangle filter and Hilbert transform. To sum up, on the basis of the above analysis, some conclusions can be obtained:

(1)  It is shown that Kasami sequence-coded UGW signals can efficiently increase the $SNR_G$ (the gain of SNR) by 6.29 dB and have a strong anti-noise performance. As the number of carrier cycles increases, the main lobe width of the decoded signal envelope gradually widens and amplitude increases. Generally, the main lobe width is mainly determined by the number of carrier cycles and the carrier waveform, and the size of the side lobes is mainly determined by the number of coding bits.

(2)  It is found that the presented adaptive peak detection algorithm has strong robustness and immunity to noise. The presented method is easy to accomplish by programming and to integrate into a real-time detection system. To improve defect positioning accuracy, the time shift resulting from bandpass filtering, triangle filtering and Hilbert transform should be small as soon as possible.

**Author Contributions:** Conceptualization, W.Y. and Y.Y.; methodology, W.Y.; validation, W.Y., Y.Y. and X.W.; writing—original draft preparation, W.Y.; visualization, W.Y.; supervision, Y.Y. All authors have read and agreed to the published version of the manuscript.

**Funding:** This research was funded by the National Natural Science Foundation of China, grant number 62101228, Youth Science and Technology Foundation of Gansu Province, China, grant number 21JR7RA245 and Shaanxi innovation capability support project, China, grant number 2021TD-25.

**Data Availability Statement:** Not applicable.

**Conflicts of Interest:** The authors declare no conflict of interest.

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
