# Peer review of "Research on Coded Excitation Using Kasami Sequence in the Long Rail Detection Based on UGW"

_electronics, doi:10.3390/electronics11091465_

Round 1

Reviewer 1 Report

Please, refer to the attached file.

Author Response

Dear Editors and Reviewers:

Thank you for your letter and for the reviewers’ comments concerning our manuscript entitled “Research on Coded Excitation Using Kasami Sequence in the Long Rail Detection Based on UGW” (No. electronics-1679612). Those comments are all valuable and very helpful for revising and improving our paper, as well as the important guiding significance to our researches. We have studied comments carefully and have made correction which we hope meet with approval. Revised portion are marked by using the “Track change” functions in Microsoft Word. The main corrections in the paper and the responds to the reviewer’s comments are illustrated as follows:

The manuscript by Yao et al. presents a “coded excitation based on Kasami sequences” to “improve SNR of the received signals and range resolution” in long rail detection based on UGWs. The research problem is well posed because increasing the signal-to-noise ratio (SNR) and range resolution in damage detection methods exploiting UGWs is a hot research topic in these years, especially if applied to structural components as railways. However, the paper is not well organized, results should be better presented, figures prepared with more care and introduction complemented.

Response: Thank you very much for your reviewing. The problems you mentioned have been solved based on your specific suggestions as shown in below.

Comment 1: Text organization and Results. The paper currently contains 19 figures, which in my opinion makes the paper too heavy to read. This is a consequence of the results that are not descried in a concise way. For instance, Figs. 6 – 8 maybe could be put together in a single figure. Some others I guess can simply be eliminated (such as Fig. 1).

Response: Thank you very much for your good suggestions. According to your suggestions, we have put together Figs. 6-8 in a single figure. In addition, Figs. 12-13 and Figs. 16-17 do the same processing as Figs. 6-8. We agree with your opinion and Fig. 1 has been eliminated in the revised paper.

Figure 5. ACF envelopes analysis of Kasami sequences, (a) (b)comparing the effect of different bits on ACF envelopes, (c) (d) comparing the effect of different cycles on ACF envelopes, (e) (f) comparing the effect of different carrier waveform.

Comment 2: Quality of the figures. I suggest improving the quality of the figures so to make numbers and text bigger and clearer to be read.

Response: We are very sorry for our negligence resulting in low quality of the figures. The numbers and text in the figures have been revised, and they are clear to be read. All figures have been modified in the revised paper.

Figure 2. PSpice model of a SPUT in longitudinal vibration.

Comment 3: Introduction. I enjoyed reading the introduction and seeing how the problem has been stated (which is a point of strength of the paper). However, I suggest complementing this section with some further citations. For instance, when talking about UGw used for damage detection in SHM, I would consider suggesting comprehensive review books, such as: “Ostachowicz et al., Guided Waves in Structures for SHM: The Time - Domain Spectral Element Method, Wiley Edition (2013)” and some concrete applications such as impact (“Miniaci et al., Application of a laser-based time reversal algorithm for impact localization in a stiffened aluminum plate, Frontiers in Materials 6, 30 (2019)”) and damage detection (“Meo et al., Detecting damage in composite material using nonlinear elastic wave spectroscopy methods. Applied Composite Materials 15(3), 115–126 (2008)” and “Ulrich et al., Interaction dynamics of elastic waves with a complex nonlinear scatterer through the use of a time reversal mirror. Physical Review Letters 98(10), 104301 (2007)”).

Response: Thank you very much for your good suggestions. These references you suggested have been added in the revised paper.

“1.   Li, J., Lu, Y., Guan, R., et al. Guided waves for debonding identification in CFRP-reinforced concrete beams. Construction and Building Materials 2017, 131, 388-399.

  1. Ostachowicz. Guided Waves in Structures for SHM: The Time - domain Spectral Element Method. Wiley Edition, 2012, ISBN: 9780470979839.
  2. Miniaci, M., Mazzotti, M., Radzienski, M., et al. Application of a Laser-Based Time Reversal Algorithm for Impact Localiza-tion in a Stiffened Aluminum Plate. Frontiers in Materials 2019, 6, 30.
  3. Meo, M., Polimeno, U., Zumpano, G. Detecting damage in composite material using nonlinear elastic wave spectroscopy methods. Applied Composite Materials 2008, 15(3), 115-126.
  4. Ulrich, T.J., Johnson, P.A., Guyer, R.A. Interaction dynamics of elastic waves with a complex nonlinear scatterer through the use of a time reversal mirror. Physical Review Letters 2007, 98(10), 104301.”

Comment 4: When talking about methods to improve the SNR, I would also consider citing passive techniques such as: “Miniaci et al., Proof of concept for an ultrasensitive technique to detect and localize sources of elastic nonlinearity using phononic crystals, Physical Review Letters 118 (21), 214301 (2017)” and “Ciampa et al., Phononic Crystal Waveguide Transducers for Nonlinear Elastic Wave Sensing, Scientific Reports 7: 14712 (2017)”.

Response: Thank you very much for your good suggestions. According to your suggestions, the two papers you share with us have been added in the references.

“In order to improve SNR of the received UGW signals, some improvements were performed from the transmitting and receiving ends. In the transmitting end, electrical impedance matching [10], optimal excitation frequency tracking [11], and driving circuit optimization [12], phononic crystals [13-14] were investigated.”

Comment 5: Also, when talking about UGw compensation procedures, I would suggest citing: “De Marchi et al., A dispersion compensation procedure to extend pulse-echo defects location to irregular waveguides, NDT & E International 54, 115-122 (2013)” and “Yang et al., Parameterised time frequency analysis methods and their engineering applications: A review of recent advances, 119, 182-221 (2019)”.

Response: We are very sorry for our negligence about the detailed explanation of the field test section.

“some methods of dispersion compensation were involved [15-17]. Xu C.B. et al. [15] proposed a guided wave dispersion compensation method by using compressed sensing technique. Marchi L.D. et al. [16] proposed a time-frequency signal processing procedure aimed at extending pulse-echo defect detection methods based on guided waves to irregular waveguides. ”

Comment 6: For these reasons, I am suggesting major revision.

Response: Thank you very much for your careful review and provide the revision opportunity.

Special thanks to you for your good comments.

We have tried our best to improve the manuscript and made some changes in the manuscript.  These changes will not influence the content and framework of the paper. And here we do not list the changes but they are marked by using Track change in revised paper. We appreciate for your warm work earnestly, and hope that the correction will meet with approval. Once again, thank you very much for your comments and suggestions.

Thank you and best regards.

Yours sincerely,

Wenqing Yao

Corresponding author: Yuan Yang

E-mail: yangyuan@xaut.edu.cn

Reviewer 2 Report

Dear authors,

the paper "Research on Coded Excitation Using Kasami Sequence in the Long Rail Detection Based on UGW” has an introduction that I enjoyed reading since it nicely went through quite some options in optimizing the detection of ultrasound signals on rails.

However, the conclusion drawn in the material section (line 84f) are in appropriate at that point of the paper. The material section should state the materials and methods and not judge on the appropriateness of the methods. This has to be moved to the discussion and conclusions.

I also did not find all symbols for equation 1 and 2 explained in the text, e.g. Z0, N, L, F. Please do explain for every symbol in every equation through out of the paper, even if you feel this not being necessary.

The unit of Z0 is not Ohm. Rho has kg/m^3, v_t has m/s, A has m^2. Thus, Z0 has kg/s. This is an acoustic and not an electric impedance!

In lines 106f Kas is written as a function of n, M and L. However, M and L are functions depending on n only. Thus, KAS is only n dependent. Hence, writing KAS(n,M,L) is irritating and mathematically not really helpful.

The approbation BPSK is not explained in the text. All appreciations should be written out at least once in the text. Do check on all appreciations you use, not only on the one that I found not being explained.

I also tried to find the Kasami sequence used in the text, however I was not success full. I would find it more than helpful if these sequences are provided in case any researcher likes to use it for their own work. I checked with reference 19. I did not find the sequence in there, but reference to other papers 51 and 52. Perhaps the authors should also read the paper they cite and check that the information they claim are written in there. I found the way how to construct the sequence in Kasami, T. (1966). Weight distribution formula for some class of cyclic codes. Coordinated Science Laboratory Report no. R-285.

I find the size of the axis labels a bit small through out all the figures. I also wonder, if one could have spared some zeros e.g. in figure 4 by using kilo-samples.

The argument in line 368 “According to equation (17), the time shift from cross-correlation is fixed for any deterministic signal.” is wrong. I cannot draw such conclusions from a MATLAB command.

In line 182 I find it very hard to draw a general conclusion by two examples only. If one really wants to make such broad general statement one would have to show this at least by a sequence of five to ten sequence of growing length. (This statement is plausible and the example shown strengthens this, but there are no reasons why this has to be generally true.)

Line 231ff it is written “According to Figure 10(a)-(d), it can be seen that with the increase of the carrier cycle number, the received signal amplitude gradually increases, but it does not increase linearly, and then the second largest peak has a larger peak increase.” I do not see this. The y-axis do have all different scales which makes comparing hard and the difference except for 2 cycles are not really super huge. This claim has to be worked out much more clearly. Do extract numbers, draw a graph with cycle numbers on the x-axis and present the prove to your claims by such a graph.

The dashed lines in figure 10(e)-(h) are not explained.

Line 256 and line 259. Here references are set to the wrong figures.

The claim line 260 should be presented by a graph that has the numbers of cycles as an x-axis and the envelope width as the y-axis.

Figure captions of  figures 16, 17, 18 are insufficient the individual graphs (a), (b), … are not explained.

I did not find the prove of the statement “It is shown that Kasami sequence coded UGW signals can efficiently increase SNR by 6 dB and have a strong anti-noise performance.” made in the conclusion being supported somewhere in the main text. From which equation of graph are the 6dB derived from?

In line 399 a line break seems to miss. “The encoding excitation under TRT and how to reduce the above time shift will be expected to perform in subsequent research work.” Is not conclusion but an outlook.

Line 405: “please add” should be removed.

By the examples give I hopefully made clear that the full reasoning and conclusion taking has to be reworked. There are quite some claims that might be true, but where no support or logic is provided why this must be so. I do ask that all statement including the ones that I did not prove to be not fully supported by the figures and equations are checked and possibly be improved. The same is true for all the references. I do really dislike when I dig into cited papers and do not find the material that is claimed to be found there.

Hence, the full paper needs a major revision and hopefully the authors correct the stuff that I did not find on first reading.

Author Response

Dear Reviewers:

Thank you very much for the reviewers’ comments concerning our manuscript entitled “Research on Coded Excitation Using Kasami Sequence in the Long Rail Detection Based on UGW” (No. electronics-1679612). Those comments are all valuable and very helpful for revising and improving our paper, as well as the important guiding significance to our researches. We have studied comments carefully and have made correction which we hope meet with approval. Revised portion are marked by using the “Track change” functions in Microsoft Word. The main corrections in the paper and the responds to the reviewer’s comments are illustrated as follows:

The paper "Research on Coded Excitation Using Kasami Sequence in the Long Rail Detection Based on UGW” has an introduction that I enjoyed reading since it nicely went through quite some options in optimizing the detection of ultrasound signals on rails.

Response: Thank you very much for your affirmation of our paper.

Comment 1: However, the conclusion drawn in the material section (line 84f) are in appropriate at that point of the paper. The material section should state the materials and methods and not judge on the appropriateness of the methods. This has to be moved to the discussion and conclusions.

Response: Thank you very much for your comments. According to your good suggestions, this has been deleted the revised paper.

Comment 2: I also did not find all symbols for equation 1 and 2 explained in the text, e.g. Z0, N, L, F. Please do explain for every symbol in every equation through out of the paper, even if you feel this not being necessary.

Response: We are so sorry for our negligence about the illustration of all symbols in equations. Now every symbol in equation 1 and 2 has been explained in the revised paper.

“Here, represents resonance frequency,  indicates the length of the transmission line,  denotes the normalized length,  represents sound velocity;  represents the characteristic impedance,  is the material density,  indicates the cross-sectional area.”

Comment 3: The unit of Z0 is not Ohm. Rho has kg/m^3, v_t has m/s, A has m^2. Thus, Z0 has kg/s. This is an acoustic and not an electric impedance!

Response: Thank you very much for your careful reviewing. From the perspective of unit, the unit of Z0 is indeed not Ohms. For the characteristic impedance, its unit is Ohm. In our paper, the equation 2 is only to conveniently calculate characteristic impedance value. The characteristic impedance and the acoustic impedance have the following relationship. For the purpose of correlating the two theories (electrical and acoustical theories), the impedance type analogy relationship is selected in which the mechanical force is denoted by the voltage and the current denotes particle velocity. The equivalence between the systems is denoted as:

Here,  represents the electrical impedance,  indicated the acoustic impedance,  represents the cross-sectional area of transmission line,  is the density and  denotes the sound speed in the material.

For the problem, more detail explanations can be found in the published paper.

[1] Wei X., Yang Y., Yao W.Q., et al. PSpice modeling of a sandwich piezoelectric ceramic ultrasonic transducer in longitudinal vibration. Sensors 2017, 17, 2253.

Comment 4: In lines 106f Kas is written as a function of n, M and L. However, M and L are functions depending on n only. Thus, KAS is only n dependent. Hence, writing KAS(n,M,L) is irritating and mathematically not really helpful.

Response: We agree with your opinion. In fact, Kas is only n dependent, but Kas(n,M,L) is used to easy identification of sequence types and lengths.

Comment 5: The approbation BPSK is not explained in the text. All appreciations should be written out at least once in the text. Do check on all appreciations you use, not only on the one that I found not being explained.

Response: We are very sorry for our negligence about the explanation of BPSK. The approbation BPSK has been explained in the revised paper.

“Because the baseband spectrum of Kasami sequences cannot satisfy the bandwidth of SPUTs used in this paper. Therefore, Kasami sequences cannot be used directly to excite a SPUT, which is the same as other codes. Then BPSK (Binary Phase Shift Keying) technique is used to modulate the Kasami sequences with a carrier signal.”

Comment 6: I also tried to find the Kasami sequence used in the text, however I was not success full. I would find it more than helpful if these sequences are provided in case any researcher likes to use it for their own work. I checked with reference 19. I did not find the sequence in there, but reference to other papers 51 and 52. Perhaps the authors should also read the paper they cite and check that the information they claim are written in there. I found the way how to construct the sequence in Kasami, T. (1966). Weight distribution formula for some class of cyclic codes. Coordinated Science Laboratory Report no. R-285.

Response: We are very sorry for our negligence about the explanation of Kasami sequences selected in the paper. The Kasami sequences used in this paper have been listed in Table 1.

“Table 1. Kasami sequences used in this paper

Bits

Types

Specific values

63

01

1       -1     1       1       -1     -1     -1     1       1        1       -1     1       -1     -1     -1     -1     1        1       -1     1       -1     1       1       1       -1     -1       1       1       1       1       -1     1       1       1        1       1       -1     -1     -1     -1     -1     -1        1       -1     1       -1     1       -1     -1     1        1       -1     -1     1       -1     -1     -1     1       -1       -1     1       -1     1

63

02

1       -1     -1     1       1       1       1       1       1       -1       -1     -1     1       1       -1     -1     -1     1        1       -1     1       1       1       -1     -1     1       -1       -1     1       1       1       1       -1     -1     -1        1       -1     1       -1     1       1       1       1       -1       -1     -1     -1     1       1       1       1       1       -1       -1     1       1       -1     1       1       -1     -1        1       -1

15

01

-1     -1     1       1       -1     1       1       1       -1     -1       -1     -1     1       -1     1

Comment 7: I find the size of the axis labels a bit small through out all the figures. I also wonder, if one could have spared some zeros e.g. in figure 4 by using kilo-samples.

Response: We are so sorry for the quality of figures. All the figures in the paper have been revised. In addition, the x axis labels have been adjusted by using kilo-samples.

“  

Figure 3. Kasami sequence analysis of ACF and CCF, (a) one 63-bit Kasami sequence, (b) the other 63-bit Kasami sequence, (c) and (d) the corresponding ACF and CCF.”

Comment 8: The argument in line 368 “According to equation (17), the time shift from cross-correlation is fixed for any deterministic signal.” is wrong. I cannot draw such conclusions from a MATLAB command.

Response: We are very so sorry for the argument. The argument has been modified in the revised paper. (See lines 397-398)

Comment 9: In line 182 I find it very hard to draw a general conclusion by two examples only. If one really wants to make such broad general statement one would have to show this at least by a sequence of five to ten sequence of growing length. (This statement is plausible and the example shown strengthens this, but there are no reasons why this has to be generally true.)

Response: Thank you very much for your good suggestions. Supplemental experiments are not possible due to the Covid-19 outbreak where I work, hence, this work will be performed in subsequent work.

Comment 10: Line 231ff it is written “According to Figure 10(a)-(d), it can be seen that with the increase of the carrier cycle number, the received signal amplitude gradually increases, but it does not increase linearly, and then the second largest peak has a larger peak increase.” I do not see this. The y-axis do have all different scales which makes comparing hard and the difference except for 2 cycles are not really super huge. This claim has to be worked out much more clearly. Do extract numbers, draw a graph with cycle numbers on the x-axis and present the prove to your claims by such a graph.

Response: Thank you very much for your suggestions. According to your suggestions, we extract numbers and draw a graph with cycle numbers on the x-axis.

“  

Figure 9. The effect analysis of carrier cycles on the amplitude of received signals and MLW, (a) 15-bit amplitude, (b) 15-bit MLW, (c) 63-bit amplitude, (d) 63-bit MLW.”

Comment 11: The dashed lines in figure 10(e)-(h) are not explained.

Response: Thank you very much for your comments. The dashed lines in figure 10(e)-(h) are used in determining MLW.

“     

  Figure 7. Results analysis of encoding excitation by 15bit Kasami sequence under different carrier cycles, (a) (b) (c) (d) the corresponding received signals, (e) (f) (g) (h) the decoding results.”

Comment 12: Line 256 and line 259. Here references are set to the wrong figures.

Response: Thank you very much for your comments. According to your suggestions, we have solved this problem in the revised paper.

“From Figures 9(a) and 9(b), with the increasing of carrier cycles, the amplitude values are gradually increased and then almost unchanged, while the MLW are gradually improved. From Figures 9(c) and 9(d), with the increasing of carrier cycles, the amplitude values first increase and then gradually decrease, while the corresponding MLW are gradually improved.”

Comment 13: The claim line 260 should be presented by a graph that has the numbers of cycles as an x-axis and the envelope width as the y-axis.

Response: Thank you very much for your comments. According to your suggestions, the graph has been drawn that has the numbers of cycles as an x-axis and the envelope width as the y-axis.

“  

Figure 9. The effect analysis of carrier cycles on the amplitude of received signals and MLW, (a) 15-bit amplitude, (b) 15-bit MLW, (c) 63-bit amplitude, (d) 63-bit MLW.”

Comment 14: Figure captions of  figures 16, 17, 18 are insufficient the individual graphs (a), (b), … are not explained.

Response: We are so sorry for the illustrations of figure captions of figures 16, 17, 18. The individual graphs (a), (b), … have been illustrated in the revised paper.

“  

Figure 13. Peak detection of the received signals under different test distances, (a) bandpass filtered signal under test distance equaling 100 m, (b) the corresponding square transformed signal, (c) the corresponding envelope, (d) peaks detection results, (e) bandpass filtered signal under test distance equaling 200 m, (f) the corresponding square transformed signal, (g) the corresponding envelope, (h) peaks detection results.

Figure 14. The analysis of peak detection under Gaussian noise, (a) SNR=-20dB, (b) peaks detection.”

Comment 15: I did not find the prove of the statement “It is shown that Kasami sequence coded UGW signals can efficiently increase SNR by 6 dB and have a strong anti-noise performance.” made in the conclusion being supported somewhere in the main text. From which equation of graph are the 6dB derived from?

Response: We are so sorry for the explanation about the “increase SNR by 6 dB”. Now the problem has been solved, the specific details are shown in below:

“The gain of SNR is defined as follows:

(8)

Here,  represents the gain of SNR,  and  indicate the amplitude of the received signals. From Figure 4(a) and Figure 4(b), it is shown that the amplitude of vibration signal in the front mass is much larger than that of single pulse, about twice as much. According to Figures 4(a) and 4(b),  and  can be obtained as 8.30 mV, 15.57 mV. Substituting these parameters into equation 8, the  can be obtained as 6.29 dB.”

Comment 16: In line 399 a line break seems to miss. “The encoding excitation under TRT and how to reduce the above time shift will be expected to perform in subsequent research work.” Is not conclusion but an outlook.

Response: Thank you very much for your suggestions. The sentence has been moved to the discussion section.

“In fact, two typical situations including transmitter-receiver-transmitter (TRT) and receiver-transmitter-receiver (RTR) are present in the real-time online detection system based on UGW. The encoding excitation under TRT and how to reduce the above time shift will be expected to perform in subsequent research work.”

Comment 17: Line 405: “please add” should be removed.

Response: We are so sorry for our negligence. According to your suggestions, “please add” has been removed in the revised paper.

Comment 18: By the examples give I hopefully made clear that the full reasoning and conclusion taking has to be reworked. There are quite some claims that might be true, but where no support or logic is provided why this must be so. I do ask that all statement including the ones that I did not prove to be not fully supported by the figures and equations are checked and possibly be improved. The same is true for all the references. I do really dislike when I dig into cited papers and do not find the material that is claimed to be found there.

Response: We are very so sorry for our negligence. Thank you very much again for your careful reviewing of our manuscript.

Comment 19: Hence, the full paper needs a major revision and hopefully the authors correct the stuff that I did not find on first reading.

Response: We have carefully revised the paper according to your good suggestions.

Thank you very much for your good comments.

We have tried our best to improve the manuscript and made some changes in the manuscript.  These changes will not influence the content and framework of the paper. And here we do not list the changes but they are marked by using Track change in revised paper. We appreciate for your warm work earnestly, and hope that the correction will meet with approval. Once again, thank you very much for your comments and suggestions.

Thank you and best regards.

Yours sincerely,

Wenqing Yao

Corresponding author: Yuan Yang

E-mail: yangyuan@xaut.edu.cn

Round 2

Reviewer 1 Report

Authors have carefully implemented my suggestions.

I endorse now its publication.

Author Response

 Thank you very much for your reviewing our manuscript again.

Reviewer 2 Report

Dear authors,

you reworked over the paper and I do appreciate that you picked up quite some of my concerns. Still there are some points left.

  1. Line 104: I fully understand why you want to have ohms there as a unit. But, it is simply wrong. You need to introduce a coefficient that transfers from acoutics to electric circuits. You should be able to find how to do this in articles and book about lumped modeling. In the end you might chose to have this transfer coefficient to be one, but one does have to be exact here. I request that there is more rigor is done here. Why did you not simply add the information that you wrote me in your response to the reviewers.
  2. Line 121: you write “It is well known …”, if it is all that clear it should be no problem to add a reference to this for all the readers to whom it is not that well known.
  3. In line 164 you introduce the signal to noise ratio gain (!) SNRG. The stated equation is only correct, if the noise is not changed. In case the noise is lowered the SNRG will be higher. Since you later elaborate on the side lope height you should include this and take this into consideration.
  4. In the conclusion you write about SNR, but it should be SNRG. Same is true for the abstract. At both places it has to be changed to SNRG.
  5. The statements in line 245ff is falsified by yourself, if you check with figure 9. The reason why I asked for that figure, was because I was unable to claims made in this paragraph. I do ask you to rework the full paragraph 243ff taking figure 9 into considerations.
  6. Figure 9 should be extended to show also the PSLs. This should make the discussion of figure 7 and 8 a lot easier. And a plot of the SNR development would also add to the quality of the discussion. It would be a lot more meaningful than the amplitudes.

Author Response

You reworked over the paper and I do appreciate that you picked up quite some of my concerns. Still there are some points left.

Response: Thank you very much for your affirmation of our revised manuscript.

Comment 1: Line 104: I fully understand why you want to have ohms there as a unit. But, it is simply wrong. You need to introduce a coefficient that transfers from acoutics to electric circuits. You should be able to find how to do this in articles and book about lumped modeling. In the end you might chose to have this transfer coefficient to be one, but one does have to be exact here. I request that there is more rigor is done here. Why did you not simply add the information that you wrote me in your response to the reviewers.

Response: Thank you very much for rigor and careful reviewing, this is very helpful to improve our paper quality.

“For the purpose of correlating the two theories (electrical and acoustical theories), the impedance type analogy relationship is selected in which the mechanical force is denoted by the voltage and the current denotes particle velocity. The equivalence between the systems is denoted as:

Here,  represents the electrical impedance,  indicated the acoustic impedance,  represents the cross-sectional area of transmission line,  is the density and  denotes the sound speed in the material.

It should be noted that for the modeling of piezoelectric transducers, the analogy method of is used. Here, by using electrical elements, the mechanical characteristics are analyzed. This is an indirect means.”

We are very sorry for the coefficient introduce, and these will be investigated in subsequent work.

Comment 2: Line 121: you write “It is well known …”, if it is all that clear it should be no problem to add a reference to this for all the readers to whom it is not that well known.

Response: We are so sorry for our negligence about the problem. According to your suggestions, the problem has been solved in the revised paper. The specific details are shown in below:

“It should be noted that good ACF is very helpful for the identification of correlation peaks at the receiver, while the low CCF values will make more effectively avoid mis-understanding other maximum values from some noises or other transmissions with the correct correlation peaks [22].”

(See lines 181-185)

Comment 3: In line 164 you introduce the signal to noise ratio gain (!) SNRG. The stated equation is only correct, if the noise is not changed. In case the noise is lowered the SNRG will be higher. Since you later elaborate on the side lope height you should include this and take this into consideration.

Response: Thank you very much for your good comments. We highly agree with your opinion. According to your suggestions, the supplementary explanation is added to the definition of SNR gain. The specific details are shown in below:

“Under the premise that the noise is determined, the gain of SNR is defined as:

(8)

Here,  represents the gain of SNR,  and  indicate the amplitude of the received signals.”

Comment 4: In the conclusion you write about SNR, but it should be SNRG. Same is true for the abstract. At both places it has to be changed to SNRG.

Response: Thank you very much for your comments. According to your suggestions, the SNR has been changed to SNRG in the abstract and conclusion. The specific details are shown in below:

“In abstract, it is shown that encoded UGW signals can increase SNRG (the gain of SNR) by 6.29 dB.

In conclusion, it is shown that Kasami sequence coded UGW signals can efficiently increase SNRG (the gain of SNR) by 6.29 dB and have a strong anti-noise performance.”

Comment 5: The statements in line 245ff is falsified by yourself, if you check with figure 9. The reason why I asked for that figure, was because I was unable to claims made in this paragraph. I do ask you to rework the full paragraph 243ff taking figure 9 into considerations.

Response: We are very sorry for our negligence about the statement. According to your suggestions, the statement has been modified in the revised paper. The specific details are shown in below:

“Based on the above experimental platform, the ultrasonic excitation signal encoded by the 15bit Kasami sequence is tested in the rail with a length of 100 m, and the number of carrier cycles is selected as 2, 4, 6 and 8, respectively. According to Figure 7(a)-(d), it can be seen that with the increase of the carrier cycle number, the received signal amplitude firstly increases, and then it is almost unchanged.

From Figure 7 and Figure 8, the values of the received signal amplitude, MLW, PSL under different bits and carrier cycles are obtained. The effect of carrier cycles on the amplitude, MLW and PSL is analyzed as shown in Figure 9. From Figures 9(a) and 9(b), with the increasing of carrier cycles, the amplitude values are gradually increased and then almost unchanged, while the MLW are gradually improved. From Figures 9(c) and 9(d), with the increasing of carrier cycles, the amplitude values first increase and then gradually decrease, while the corresponding MLW are gradually improved. Combined Figure9(e) and Figure9(f), it is found that with the increasing of carrier cycles, PSL presents minor changes. To summarize, the MLW is closely related with carrier cycles when Kasami sequences bits are unchanged, and the number of carrier cycles has little effect on PSL.”

Comment 6: Figure 9 should be extended to show also the PSLs. This should make the discussion of figure 7 and 8 a lot easier. And a plot of the SNR development would also add to the quality of the discussion. It would be a lot more meaningful than the amplitudes.

Response: Thank you very much for your good comments. According to your suggestions, Figure 9 has been extended to shown also PSLs in the revised paper. The specific details are shown in below:

“  

Figure 9. The effect analysis of carrier cycles on the amplitude of received signals and MLW, (a) 15-bit amplitude, (b) 15-bit MLW, (c) 63-bit amplitude, (d) 63-bit MLW, (e) 15-bit PSL, (f) 63-bit PSL.”

Round 3

Reviewer 2 Report

Thank you for taking my comments into considerations. From my point of view the paper is ready for publication.

I wish you much success with your future research.